# A habenula-insular circuit encodes the willingness to act

Nima Khalighinejad [1,4 ✉], Neil Garrett [1,3,4], Luke Priestley[1,4], Patricia Lockwood [1,2] & Matthew F. S. Rushworth [1]

The decision that it is worth doing something rather than nothing is a core yet understudied feature of voluntary behaviour. Here we study "willingness to act", the probability of making a response given the context. Human volunteers encountered opportunities to make effortful actions in order to receive rewards, while watching a movie inside a 7 T MRI scanner. Reward and other context features determined willingness-to-act. Activity in the habenula tracked trial-by-trial variation in participants' willingness-to-act. The anterior insula encoded individual environment features that determined this willingness. We identify a multi-layered network in which contextual information is encoded in the anterior insula, converges on the habenula, and is then transmitted to the supplementary motor area, where the decision is made to either act or refrain from acting via the nigrostriatal pathway.

[1] Wellcome Centre for Integrative Neuroimaging, Department of Experimental Psychology, University of Oxford, Oxford OX1 3SR, UK. [2] Centre for Human Brain Health, School of Psychology, University of Birmingham, Birmingham, UK. [3] Present address: School of Psychology, University of East Anglia, Norwich, UK. [4] These authors contributed equally: Nima Khalighinejad, Neil Garrett, Luke Priestley. ✉email: nima.khalighinejad@psy.ox.ac.uk

When performing a voluntary action, one has to decide not only which action to choose but whether or not it is worth initiating an action in the first place, given the potential benefits of acting in a particular environment. Consider someone weighing up whether to go to the trouble of filling out an application for a job or continue watching Netflix in the expectation that better job opportunities will eventually come along. Motivation to act requires considering both the potential reward(s) in the context of the distribution of other opportunities in the environment and the cost of action. The person may decide to act and make the application if the job is a particularly good one, but they may do so also if other job opportunities are currently few and far between or if the application form is brief and simple and requires little effort to complete. If the costs, benefits, or environment are otherwise, however, then no action may be initiated, and the would-be job seeker may remain inert and inactive.

Whilst the neural mechanisms underlying decisions about what to do and which action to select have been extensively studied[1–4], the processes underlying decisions about if and when to start an action have attracted less attention[5,6]. Given the important role these kinds of decisions play in both human and animal survival, understanding the underlying brain mechanisms is essential for building a comprehensive picture of decision making. In doing so, we can also seek to understand maladaptive behaviours that potentially arise from dysfunction in this circuitry such as the failure to adapt one's willingness-to-act when an environment deteriorates[7,8]. Such failures characterise apathy and impulsivity, symptoms prevalent in a number of psychiatric and neurological disorders[9,10].

Previous studies have emphasised the role of frontal cortical brain regions in motivated behaviour—including ventromedial prefrontal cortex (vmPFC)[11], anterior insula (AI), dorsal anterior cingulate cortex (dACC), and subgenual anterior cingulate cortex (sgACC)[12]. Other studies have focussed on specific subcortical structures that enhance motivational drive. Specifically, the nucleus accumbens (NAc), laterodorsal tegmentum (LDT)[13], ventral pallidum (VP)[14], ventral tegmental area (VTA), habenula (HB), and pedunculopontine nucleus (PPN)[15]. However, it remains unclear how these cortico-subcortical structures communicate across a distributed circuit to decide whether or not to initiate a voluntary action, given the costs and benefits of a current opportunity and the environment in which it occurs. Here, we used ultra-high field imaging (7 T fMRI) to investigate how one's environmental context influences willingness to initiate a volitional action and how it exerts this influence via brain circuits.

We have recently shown in humans that in addition to activity in cortical regions such as the anterior cingulate cortex, the earliest neural activity predicting the occurrence of a self-initiated action appeared in a group of subcortical structures including the dorsal and ventral striatum, substantia nigra (SN), basal forebrain (BF), HB, and PPN[16]. BF, in particular, mediated the influence of environmental context on the emergence of a decision about when to act[16,17]. In those studies, however, subjects had to make a response at every trial and therefore we could not investigate how, before deciding when to make an action, one decides whether or not to initiate an action in the first place. Here, by using an experimental design in which the key decision is whether to execute an action or not (rather than when an action should be executed), we set out to characterise the multi-layered circuit recruited in a related decision: whether to execute or withhold an action, given the potential benefit of acting in a particular environment.

To answer this question, we designed a task in which participants were free simply to do nothing while inside a 7 T scanner; they could simply lie still and watch a movie. However, they were also given a series of opportunities for action and they could decide whether they were willing to take these actions. If they did then they engaged in an effortful task for a potential reward (Fig. 1). Here, we show that cost and benefits of reward opportunities in a given environment influence participants' willingness to act and undertake effort in return for potential reward. Ultra-high field functional imaging shows that BOLD activity in HB is correlated with participants' willingness-to-act. In parallel to HB, anterior insula also tracks participants' willingness-to-act, but in addition encodes individual parameters of the environment that determine this willingness. Finally, using psychophysiological interaction analysis (PPI) and structural equation modelling (SEM) we identify a cortico-subcortico-cortical circuit starting in the anterior insula, converging in the habenula and ending in the supplementary motor area (SMA) via the nigrostriatal pathway.

## Results

**Contextual factors influence participants' willingness to act.** Participants ($N = 25$) watched a movie whilst inside a 7 T scanner (one was later excluded from the behavioural and brain analysis and two from brain analysis; see "Methods"). On each trial (as the movie played), participants were presented with opportunities (offers) to act in an effortful task in return for a potential reward. Offers appeared on the screen, superimposed on the movie (Fig. 1a). Participants could choose to act by pressing a button on a response pad in order to receive the benefits of the opportunity and incur its costs. If they made no response, then they simply continued to do nothing. Offer stimuli consisted in centrally presented vertical rectangles which contained small circular dots. The colour and the number of dots represented the magnitude and the probability of the potential reward, respectively (Fig. 1b). The reward magnitude and reward probability varied from trial-to-trial in a pseudorandomised order. In addition, the average value of the environment changed every block of 36 trials. Rich blocks contained a greater frequency of high reward magnitude and high probability offers, and a lower frequency of low reward and low probability offers compared to poor blocks (Fig. 1b). Participants were explicitly told at the start of each block whether they were entering a rich or a poor block (but were not told the distribution of reward magnitudes and probabilities associated with the offers in the respective blocks). In summary, we experimentally manipulated three factors in order to alter participants' environments and influence their decision: reward magnitude, reward probability and block type. Our focus was on establishing the factors that lead people to act and so we carefully quantified those factors that would lead them to act such as the potential reward benefits, probabilities and the costs. Importantly, at the same time there was no relationship between movie scenes and the task parameters and therefore interest in movie scenes could not have any systematic influence on participants' decisions (also see Supplementary Fig. S1 for formal assessment of the relationship between interest in movie scenes and participants' decisions).

If participants chose to accept an offer, by making a response, the movie was interrupted, and they performed a short effortful task which involved exerting a bout of physical effort—thresholded to their own grip strength—by squeezing a dynamometer. After the effortful task, the movie resumed, and the reward outcome was displayed on the screen. Participants received a monetary reward with the probability and at the magnitude determined by the accepted offer, if successful at performing the effort-task. They received no reward if they accepted an offer but were unsuccessful at completing the effort-task, if they were unlucky in the lottery, or if they decided not to make a response and continued watching the movie uninterrupted (see "Methods").

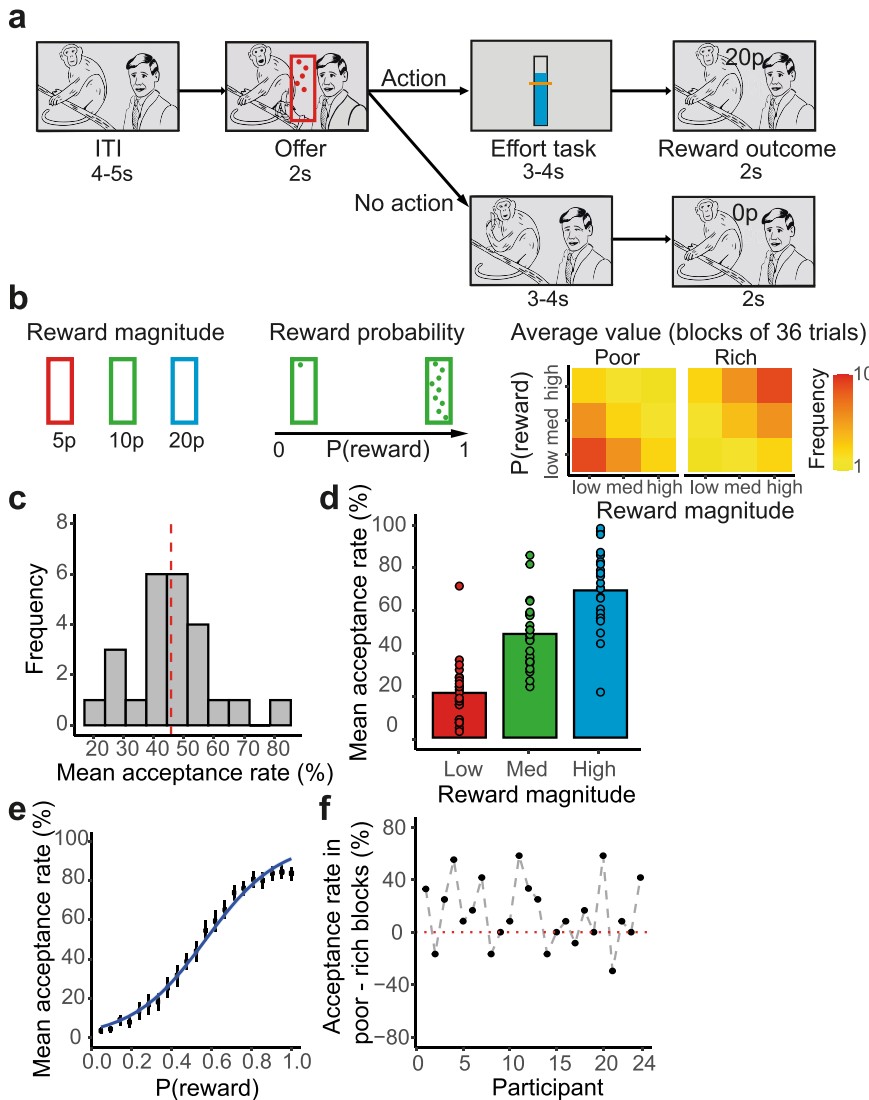

**Fig. 1 Experimental task and the main behavioural results. a** While watching a movie, participants received a series of offers in the form of visual stimuli that were superimposed upon the movie. Each offer appeared for 2 s and could be accepted by button-pad response while it remained on-screen. Offer stimuli consisted in centrally presented vertical rectangles which contained small circular dots. The colour of the offer represented reward-magnitude and the number of dots indicated the reward-probability. If the participant made a response to accept an offer (action), the movie was interrupted, and the participant needed to complete a short effort-task for a duration of 3−4 s. If the force was successfully exerted, the participant became eligible for a monetary reward of the probability and magnitude indicated by the accepted offer. The movie resumed once the effort-task was completed, and feedback about the reward outcome was superimposed on-screen for 2 s. If the offer was not accepted (no-action), participants continued to watch the movie uninterrupted for a duration equivalent to the effort-task but would receive no monetary reward. The next offer appeared after an inter-trial-interval (ITI) of 4−5 s. **b** We asked whether contextual factors in participants' environments influenced their decisions to act (i.e., to accept an offer by making a response). These factors included: (1) Offer reward magnitude with three levels: low (5p), medium (10p) or high (20p), indicated by red, green or blue (magnitude-to-colour contingencies were counterbalanced across participants). (2) Reward probability, which increased linearly with the number of dots comprising the stimulus. (3) Average value of the environment which was manipulated by changing the ratio of high magnitude and high probability offers within blocks. **c** The distribution of response rate (act/no-act decisions). The red dashed line is the average response rate across participants. **d**−**f** Participants ($n = 24$) were more likely to act when offered a large magnitude reward (**d**), a high probability reward (**e**), and when they were in a poor compared to a rich block (**f**). In panel (**d**) each ring represents one participant, and the columns illustrate the group mean. In panel (**e**) the black points and the error bars are mean response rate and standard error of the mean at each discrete level of reward-probability, respectively, across participants. Panel (**f**) shows the difference in acceptance rate of similar offers (medium magnitude and medium probability) between poor and rich blocks, for each participant. Source data are provided as a Source Data file.

We first investigated whether the various contextual factors in participants' environments including reward magnitude, reward probability, and block type influenced their decision to act (i.e., accept an offer). On average, participants chose to act for $45.81 \pm 2.79\%$ of the offers presented (Fig. 1c). A generalised linear mixed-effect model (see "Methods"; also see Supplementary Table S1) showed that participants were more likely to act

when offered higher magnitude rewards ($\beta = 2.07 \pm 0.26$, X2(1) = 48.64, $P < 0.001$; Fig. 1d), higher probability rewards ($\beta = 3.53 \pm 0.26$, X2(1) = 167.46, $P < 0.001$; Fig. 1e), and when they were in a poor compared to a rich block ($\beta = -0.85 \pm 0.18$, X2(1) = 16.33, $P < 0.001$; Fig. 1f). In addition, we asked whether the expected value (reward magnitude × probability) from the previous trial and response history (act/no-act decisions on the

previous trial) influenced participants' responses on the current trial. Participants were more likely to refrain from action the greater the expected value of the offer on the previous trial ($\beta = -0.16 \pm 0.08$, X2(1) = 3.40, $P = 0.065$). Although not statistically significant, the direction of this effect is consistent with the effect we observe for block type and suggests that participants become more selective following evidence (via trial-to-trial changes in the quality of offers presented) that their environment is becoming richer and, by contrast, become less picky when the environment gets poorer. We found no significant effect of response history ($\beta = -0.06 \pm 0.17$, $P = 0.70$), suggesting that fatigue from performing the effortful task on the previous trial did not influence participants' response on the current trial.

Overall, these findings suggest that participants took a variety of contextual factors into account when deciding whether it was worth acting in order to receive the potential reward given the level of effort that was to be exerted and given the rate at which reward opportunities occurred in a given environment. Next, we used the combined effect of these contextual factors—separately estimated for each individual participant—to infer participants' "willingness to act" on each trial. Specifically, willingness-to-act was defined as the probability of acting, predicted by the combination of opportunity and environment in a given context

(see "Methods" for full model specification). This trial-by-trial variable was then used as a parametric regressor in the model-based fMRI analyses.

**Habenula tracks participants' willingness to act.** We previously showed that activity in a subcortical network consisting of dorsal striatum (DS; including caudate nucleus and putamen), NAc, midbrain dopaminergic (MidD) system (including SN and VTA), PPN, HB, and BF (including septal nuclei and diagonal band of Broca), predicted the timing of voluntary actions (when to act)[16]. Here we selected the same set of structures as a priori regions of interest (ROIs) to ask whether the same network is involved in computing a key antecedent decision: whether to initiate an action, in the first place.

First, we investigated whether any of these structures encoded participants' observed act/no-act decisions (i.e., whether or not they acted to accept an offer; GLM2.1; Fig. 2). BOLD signals in DS (one-sample $t$ test; $t(21) = 9.78$, $P < 0.001$, $d = 2.08$; all subsequent tests are corrected for multiple comparisons), MidD ($t(21) = -3.52$, $P = 0.008$, $d = 0.75$), PPN ($t(21) = -3.09$, $P = 0.01$, $d = 0.66$), HB ($t(21) = 3.16$, $P = 0.01$, $d = 0.67$), and BF ($t(21) = -3.91$, $P = 0.004$, $d = 0.83$) were correlated with

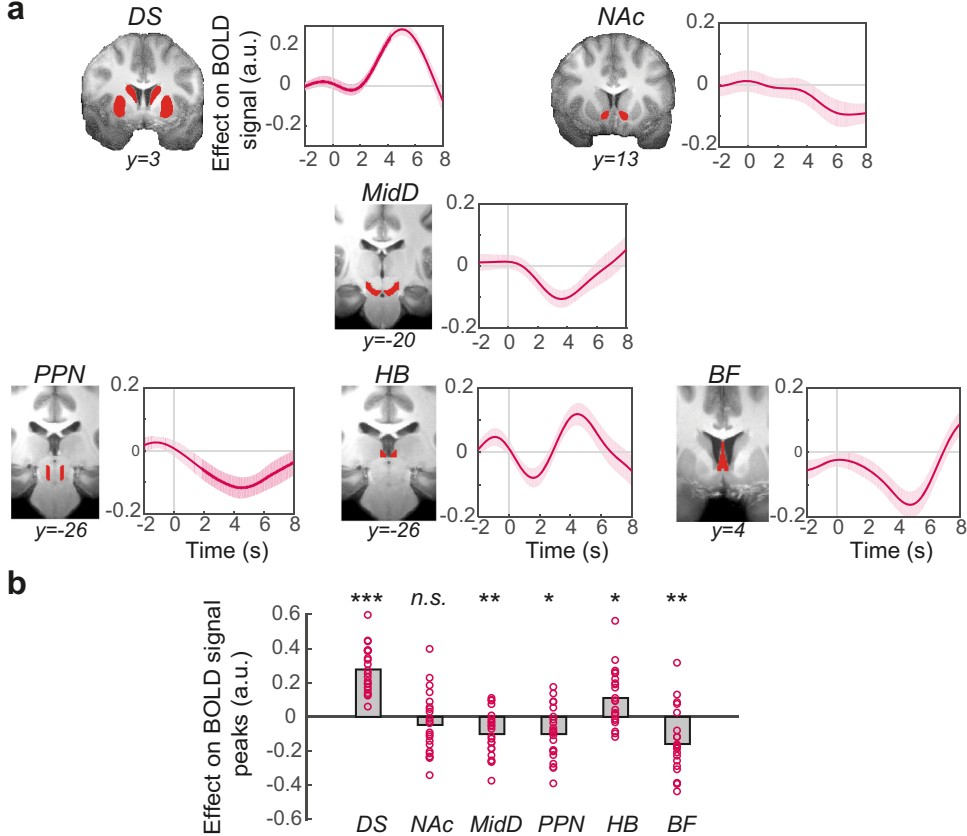

**Fig. 2 A subset of subcortical structures encodes the act/no-act decision. a** ROI time-course analysis of the a priori selected subcortical structures, showing the relationship between BOLD and act/no-act decision. The panel next to each time-course shows the corresponding anatomical ROI overlaid on averaged structural image of all subjects in standard space. The $y$ axis is based on the FSL MNI152 standard brain in which $y = 0$ is the dorsal posterior corner of the anterior commissure (ac). The lines and shadings show the mean and standard error of the $\beta$ weights across the participants, respectively. Time zero is the trial onset (appearance of the offer). Note that the hemodynamic lag means that a BOLD signal change reflects neural activity approximately 6 s earlier. DS dorsal striatum, NAc nucleus accumbens, MidD midbrain dopaminergic system, PPN pedunculopontine nucleus, HB habenula, BF basal forebrain. **b** There was a significant relationship between BOLD activity and the act/no-act decision in DS ($P = 1.7E-08$), MidD ($P = 0.008$), PPN ($P = 0.01$), HB ($P = 0.01$), and BF ($P = 0.004$). Each ring represents one participant ($n = 22$). The grey columns illustrate the group mean. Significance testing on time-course data was performed by using a leave-one-out procedure on the group peak signal. Two-sided, one-sample $t$ tests with Holm −Bonferroni correction. a.u. arbitrary units. *$P < 0.05$, **$P < 0.01$, ***$P < 0.001$, n.s. not significant. See also Supplementary Fig. S2 for a related analysis.

participants' observed decisions (action vs. inaction). This was not the case in a more ventral part of the striatum, NAc ($t(21) = -1.22$, $P = 0.24$) (Fig. 2). The same set of structures were associated with the initiation of self-paced actions in a previous study[16]. Here, however, we found a negative relationship between act/no-act decisions and BOLD activity in MidD (see Fig. 2a). This might at first seem counterintuitive given that one would expect MidD to positively encode the act/no-act decisions. The key point to note, however, is that we are considering activity that is time-locked to the stimulus onset rather than movement onset. When, by contrast, we focus on activity time-locked to movement onset and carefully examine activity in the two subsections of MidD—the substantia nigra pars compacta (SNc) and the ventral tegmental area (VTA)—we see clear evidence of activity related to movement onset in SNc as might have been expected given that many researchers[18], including ourselves[16], have previously identified SNc with action initiation (Supplementary Fig. S2).

Next, we asked whether the same set of structures tracked trial-by-trial variation in participants' willingness-to-act—that is, the probability of acting for a given opportunity in a given environment—while controlling for participants' actual observed act/no-act decisions (GLM2.1). We found that trial-by-trial variation in participants' willingness-to-act explained BOLD activity in HB ($t(21) = 4.26$, $P = 0.002$, $d = 0.90$; corrected for multiple comparisons) (Fig. 3; also see Supplementary Fig. S3 for alternative HB mask). This relationship was not found in any other ROI (all $P > 0.12$; also see Supplementary Table S2 for Bayesian analysis). This suggests that independent of the final decision whether to act or not, HB tracks the trial-by-trial variation in participants' willingness-to-act given the combination of opportunity and environment in a given context.

Here we showed a positive relationship between HB activity and the willingness-to-act, as signalled by the combination of contextual factors. Previous research, however, has shown a link between HB and control of impulsive behaviour[19]. To further investigate these two (seemingly contradictory) findings, we separately assessed the relationship between willingness-to-act and HB BOLD signal on trials where participants made an action and those in which they withheld an action in favour of continuing to watch the movie (Supplementary Fig. S4). The relationship between BOLD signal in HB and willingness-to-act was stronger in the latter compared to the former (paired-sample $t$ test; $t(21) = -4.93$, $P < 0.001$, $d = 1.05$). This suggests that an increase in willingness-to-act is associated with a strong HB BOLD signal when participants received high value offers but nevertheless decided not to act, consistent with a potential role of HB in impulse control and suppression of action, when active, and release of action when less active.

**Anterior insula mediates the influence of contextual factors on willingness to act.** Selecting ROIs a priori, combined with ultra-high field imaging, enabled us to examine neural correlates of willingness-to-act in small subcortical structures that are difficult to investigate using conventional neuroimaging methods. We next ran a whole-brain analysis to identify other potential structures—not limited to our a priori ROIs—that encoded parametric variation in willingness-to-act whilst controlling for the nature of the final decision (act or non-act) alongside other potential confounds (see "Methods", GLM1). This revealed two significant clusters; the first located in the anterior insula (peak $Z = 5.16$, MNI: $x = 43$, $y = 16$, $z = 1$; whole-brain cluster-based correction, $Z > 3.1$, $P < 0.001$; Fig. 4a), and the second located in the SMA extending into the caudal part of the anterior cingulate (peak $Z = 5.50$, MNI: $x = -6$, $y = -2$, $z = 54$; whole-brain cluster-based correction, $Z > 3.1$, $P < 0.001$; see Supplementary Table S3 for the list of clusters).

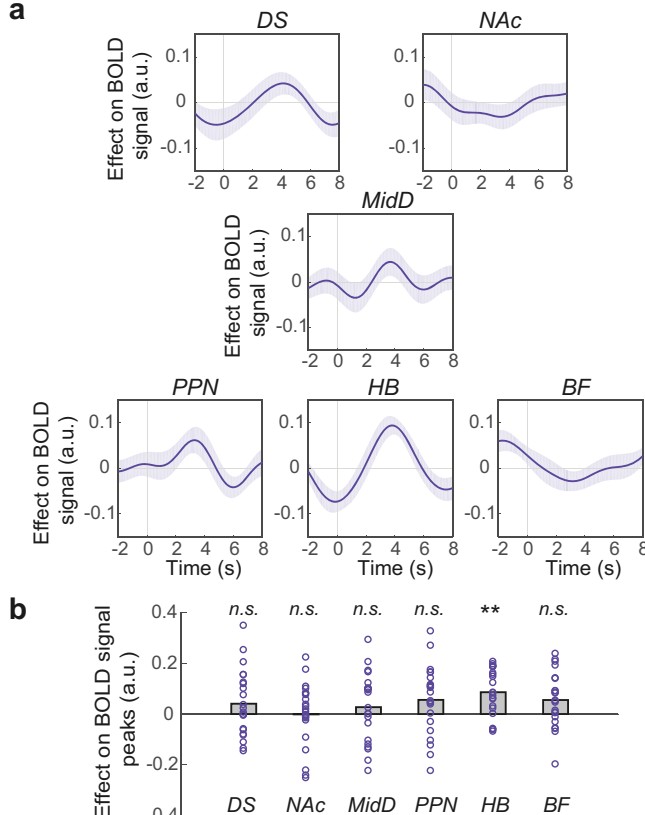

**Fig. 3 HB encodes willingness-to-act. a** ROI time-course analysis showing the relationship between BOLD activity and willingness-to-act. Format as in Fig. 2a. The lines and shadings show the mean and standard error of the $\beta$ weights across the participants, respectively. Time zero is the trial onset (appearance of the offer). Note that the hemodynamic lag means that a BOLD signal change reflects neural activity approximately 6 s earlier. **b** There was a significant relationship between BOLD activity in HB and willingness-to-act ($P = 0.002$). Each ring represents one participant ($n = 22$). Significance testing on time-course data was performed by using a leave-one-out procedure on the group peak signal. Two-sided, one-sample $t$ tests with Holm−Bonferroni correction. Format as in Fig. 2b. **$P < 0.01$, n.s. not significant, a.u. arbitrary units. See also Supplementary Fig. S3 and Supplementary Table S2.

Next, we searched for voxels in which activity was correlated not with the participants' willingness-to-act but with the translation of this willingness into actual, binary decisions of whether to act or not act (response contrast in GLM1). There was a large and significant cluster at SMA extending into the caudal part of the anterior cingulate, overlapping with the previously observed cluster for willingness-to-act (Supplementary Fig. S5). However, voxels located in the anterior insula were not correlated with whether or not participants finally decided to act or refrain from acting (compare Fig. 4a and Supplementary Fig. S5).

The whole-brain analysis thus showed a significant relationship between activity in the anterior insula and willingness-to-act, but the relationship between activity in SMA, willingness-to-act, and participants' observed act/no-act decisions is less clear. To better understand this relationship we extracted the time course of the BOLD signal from 14 mm$^3$ sphere ROIs centred on the centroid of the anterior insula and SMA activation peaks and compared the effect of willingness-to-act and the act/no-act decision on their respective BOLD signals, using the same approach as in previous time-course analysis (see "Methods", GLM2.1; Fig. 4b, c). In this approach, regressors are fit at each time step of the

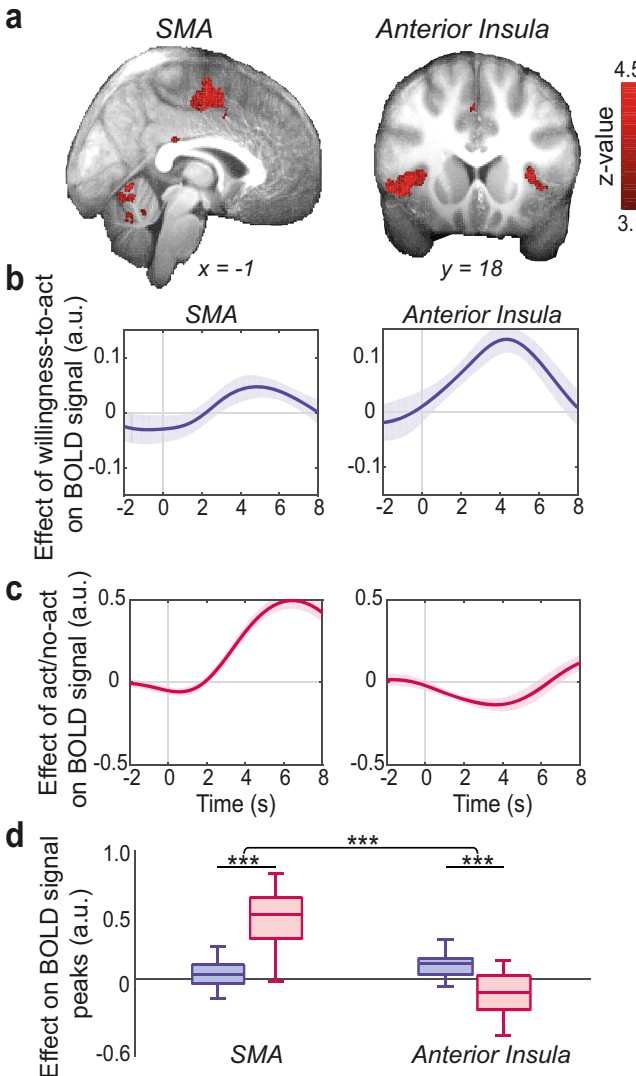

**Fig. 4 Act/no-act decisions and willingness-to-act are encoded differently in SMA and anterior insula. a** Whole-brain analysis showing voxels where activity reflected parametric variation in willingness-to-act. We focussed on two clusters; the first located in supplementary motor area (SMA) extending into the caudal part of the anterior cingulate (left panel) and the second in anterior insula (right panel). Whole-brain cluster-based correction, $Z > 3.1$, overlaid on averaged structural image of all subjects in standard space (see Supplementary Table S3 for the list of clusters). **b** ROI time-course analysis of the SMA (left panel) and anterior insula (right panel), showing the relationship between BOLD signal and willingness-to-act. **c** ROI time-course analysis of the SMA (left panel) and anterior insula (right panel), showing the relationship between BOLD signal and the act/no-act decision. Time zero is the response time. Note that the hemodynamic lag means that a BOLD signal change reflects neural activity approximately 6 s earlier. In (**b**) and (**c**) the lines and shadings show the mean and standard error of the $\beta$ weights across the participants, respectively. **d** BOLD signal in anterior insula had a stronger positive correlation with willingness-to-act (blue boxes) compared to act/no-act decisions (red boxes; $P = 0.0004$). BOLD signal in SMA had a stronger positive correlation with act/no-act decisions compared to willingness-to-act ($P = 2.29E{-}08$; interaction effect: $P = 8.15E{-}09$). In box plots, the central line indicates the median and the bottom and top edges of the box indicate the 25th and 75th percentiles, respectively. Whiskers extend to the most extreme data points not considered outliers. $n = 22$ participants. Significance testing on time-course data was performed by using a leave-one-out procedure on the group peak signal. Two-way repeated-measures ANOVA followed by paired-samples $t$ test. ***$P < 0.001$. a.u. arbitrary units.

epoched data and are therefore better suited to dissociate the effect of willingness-to-act from the act/no-act decision on SMA and the anterior insula. Whilst willingness-to-act better explained BOLD signal in the anterior insula compared to SMA (paired-sample t-test on peak signal; $t(21) = 3.01$, $P = 0.007$, $d = 0.64$; Fig. 4b), participants' act/no-act decisions better explained BOLD signal in SMA compared to anterior insula (paired-sample $t$ test on peak signal; $t(21) = 10.42$, $P < 0.001$, $d = 2.22$; Fig. 4c; two-way repeated-measures ANOVA; $F(1,21) = 84.66$, $P < 0.001$, $\eta_p^2 = 0.80$; Fig. 4d). This suggests that at the onset of the circuit, anterior insula–in parallel to HB—encodes willingness-to-act given the value of current opportunities in the current environment, whilst SMA, translates the willingness-to-act into an actual action at a later stage in the circuit (also note in Fig. 4 the later peak of the act/no-act effect on SMA compared to willingness-to-act effect on anterior insula). This is consistent with previous reports associating SMA with planning and execution of voluntary actions[20,21].

**Anterior insula tracks individual component features of the environment.** HB and anterior insula both encoded willingness-to-act. The presence of such an activity pattern could be for one of two reasons. It could arise because the area tracks each component factor relating to the current opportunity in the current

environment such as the reward magnitude, reward probability, whether the current block is rich or poor, and other aspects of recent experience. Alternatively, it could arise because each individual component factor is encoded elsewhere and only their integrated impact on willingness-to-act is encoded in the brain area. To arbitrate between these hypotheses, we used a new model to investigate the distinct effect of each contextual factor—reward magnitude, reward probability, block-type, and the expected value of the previous trial—on BOLD signal (GLM2.2; Fig. 5a). Trial-by-trial variation in each component factor was correlated with BOLD signal in the anterior insula (Fig. 5b). Interestingly, the direction of the impact of each component factor on anterior insula activity paralleled the impact that each component factor had on behaviour (except for the expected value): positive deflection in BOLD signal was associated with higher magnitude rewards (one-sample $t$ test; $t(21) = 2.83$, $P = 0.03$, $d = 0.60$; all subsequent tests are corrected for multiple comparisons), higher probability rewards ($t(21) = 6.02$, $P < 0.001$, $d = 1.28$), being in a poor compared to a rich block ($t(21) = 2.83$, $P = 0.03$, $d = 0.60$), and greater expected value of the offer on the previous trial ($t(21) = 2.73$, $P = 0.02$, $d = 0.58$). This was not the case in HB, in which reward magnitude was the only component factor that, even by itself, correlated with BOLD signal ($t(21) = 3.71$, $P = 0.005$, $d = 0.79$). This shows that the reward features of the current environment that determine willingness-to-act are also associated with variation in activity in the neural structures that encode willingness-to-act. While HB encodes a simple and directly cued feature of the reward environment—the reward opportunity potentially available right now—the cortical component of the circuit—anterior insula—encodes features of the reward environment in a more complex way—it encodes not just the reward opportunity available right now but in addition it encodes the sparseness/richness of the environment in which that opportunity occurs. However, given that both HB and anterior insula eventually come to encode willingness-to-act, this suggests that contextual information may subsequently be passed to HB to

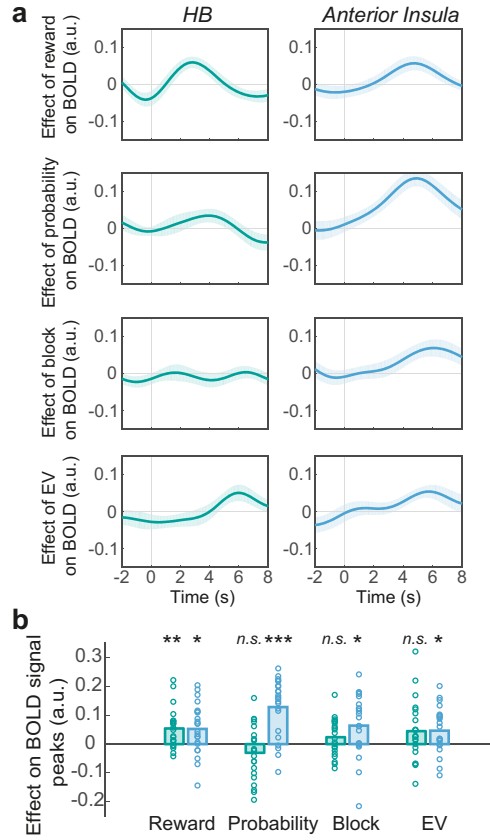

**Fig. 5 Anterior insula tracks individual component features of the environment. a** ROI time-course analysis showing the relationship between BOLD activity in habenula (HB) (left side panels) and anterior insula (right side panels) with reward magnitude, reward probability and block type (poor vs. rich) on the current trial, and the expected value (EV) of the offer on the previous trial. Format as in Fig. 2a. The lines and shadings show the mean and standard error of the β weights across the participants, respectively. Time zero is the trial onset (appearance of the offer). Note that the hemodynamic lag means that a BOLD signal change reflects neural activity approximately 6 s earlier. **b** There was a significant relationship between BOLD activity in insula (blue boxes) and all four contextual factors (Reward, $P = 0.03$; Probability, $P = 2.25E−05$; Block, $P = 0.03$; EV, $P = 0.02$). In HB (green boxes) reward magnitude was the only component factor that correlated with BOLD signal ($P = 0.005$). Each ring represents one participant ($n = 22$). Significance testing on time-course data was performed by using a leave-one-out procedure on the group peak signal. Two-sided, one-sample $t$ tests with Holm−Bonferroni correction. *$P < 0.05$, **$P < 0.01$, ***$P < 0.001$, n.s. not significant, a.u. arbitrary units.

guide the decision about whether or not to act. We test this hypothesis in the next section.

**Willingness to act is constructed across a multi-layered network.** Our results so far suggest that willingness-to-act is constructed within an anterior insula-HB circuit, in which individual component determinants of willingness-to-act are tracked by anterior insula, integrated and may then be passed to HB. This implies a functional coupling between the two structures, driven by the parametric variation in willingness-to-act. To test this hypothesis, we performed a PPI analysis between HB and anterior insula BOLD signals with willingness-to-act as the psychological factor (see "Methods"; GLM2.3). As predicted, parametric variation in willingness-to-act influenced the relationship between HB and anterior insula (one-sample $t$ test; $t(21) = −2.29$, $P = 0.03$, $d = 0.49$). This was not the case for the functional

connection between HB and SMA ($P = 0.61$). This suggests that the anterior insula is functionally connected with HB and that this connection is moderated by participants' willingness-to-act.

However, for a response to happen, the willingness to initiate an action needs to be translated into actual action. HB sends direct projections to dopaminergic midbrain (MidD)[19,22,23], and the role of MidD, particularly the substantial nigra pars compacta (SNc) in action initiation per se was highlighted in our earlier analyses (Fig. 2 and Supplementary Fig. S2a) and in previous studies[5,16,18]. We therefore asked whether the interaction between willingness-to-act and the act/no-act decision influenced the relationship between HB and MidD. PPI analysis (see "Methods"; GLM2.4) showed that this was the case; BOLD signal in HB was correlated with BOLD signal in MidD and this correlation was moderated by the interaction of willingness-to-act and the act/no-act decision ($t(21) = −2.09$, $P = 0.049$, $d = 0.44$). As previously (Supplementary Fig. S2), we next considered the possibility that MidD subdivisions, SNc or VTA may make different contributions to the translation from willingness-to-act to actual action initiation. We therefore, once again, used the VTA and SNc specific ROIs[24] to perform this analysis. The same PPI effect was significant between HB and SNc ($t(21) = −2.56$, $P = 0.018$, $d = 0.55$) but not between HB and VTA ($t(21) = 1.49$, $P = 0.15$), and, moreover, this relationship was stronger in the former compared to the latter pathway (paired-sample $t$ test; $t(21) = 3.36$, $P = 0.003$, $d = 0.72$). In summary, the results discussed here and in Fig. 3 and Supplementary Fig. S2 reveal that HB encodes willingness-to-act at the time that an action cue first appears although SNc does not. However, activity coupling begins to occur between HB and SNc as a function of willingness to act, and then an activity increase occurs in SNc that is time-locked to the moment of movement onset.

More generally, the findings from the various PPI analyses are suggestive of a cortico-subcortical-cortical functional network in which willingness-to-act emerges within an anterior insula-HB circuit and is passed on to dopaminergic midbrain to influence the actual act/no-act decision via the nigrostriatal pathway. These observations suggest a functional model (Fig. 6a) in which anterior insula influenced HB, in line with the results of our first PPI analysis. The model reflected the observation in the second PPI analysis that HB influenced MidD. The model also incorporated our previous finding that activity in other structures—BF and PPN—also influence SNc[16], although both that previous study and the current one suggest that they convey a distinct type of information to SNc. The model is also concordant with the anatomical projections from BF, PPN, and HB to MidD that have been reported[25–28]. Finally, we assumed projections back into the cortex (SMA) via the nigrostriatal pathway (Fig. 6a). Anatomical connections projecting from MidD to ventral and dorsal striatum and the influence MidD exerts on them are well-known[29]. To formally test the plausibility of this model we used structural equation modelling (SEM). SEM—a well-established method for analysing functional and effective connectivity[16,30,31]—probes the direction and strength of connections between our ROIs, rather than activity within individual ROIs (see "Methods").

First, we estimated the path coefficients to determine whether the interrelationship between the BOLD signals of our ROIs fit the proposed model. As predicted, all specified path coefficients in the hypothesised model were significantly different from zero (Supplementary Table S4). Next, we asked how well the hypothesised model (Model 1) described the data relative to alternative but plausible models of similar complexity. Model 2 was similar to Model 1 but with opposite direction of information flow (Fig. 6b). Model 3 was similar to Model 1 but assumed the opposite pattern of key cortico-subcortical links; it assumed that

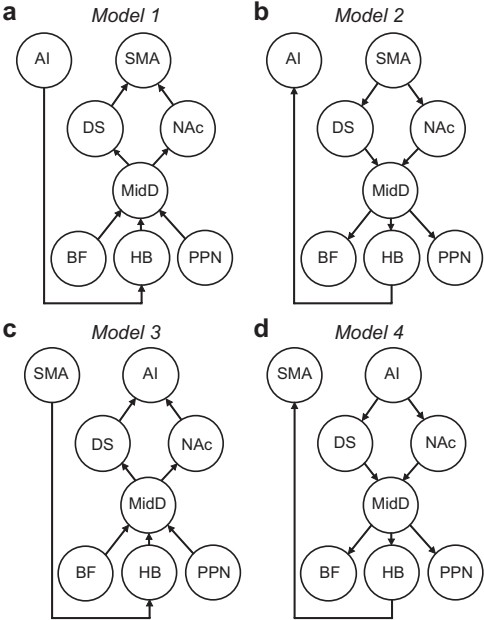

**Fig. 6 Willingness-to-act emerges across a multi-layered network. a** The hypothesised functional model in which activity in the supplementary motor area (SMA) is influenced by dorsal striatum (DS) and nucleus accumbens (NAc). Activity in DS and NAc is, in turn, influenced by dopaminergic midbrain (MidD). Activity in MidD is influenced by basal forebrain (BF), pedunculopontine nucleus (PPN) and habenula (HB); and anterior insula (AI) influences HB (for estimates of path coefficients, see Supplementary Table S4). **b−d** Alternative models. The hypothesis-driven model (Model 1) fits the data better than the alternative models (Models 2−4).

SMA, rather than anterior insula, influenced HB and that anterior insula, rather than SMA, was influenced by the nigrostriatal pathway (Fig. 6c). Finally, Model 4 was similar to Model 3 but with opposite direction of information flow (Fig. 6d). Model comparison showed that the hypothesised model (Model 1) performed better than all alternative models (AIC; Model 1 = 376,817; Model 2 = 534,500; Model 3 = 377,651; Model 4 = 534,141). This suggests that decisions about whether or not to initiate an action emerge within a cortico-subcortical-cortical functional network, starting in an anterior insula-HB pathway and ending in SMA via the nigrostriatal pathway.

## Discussion

An important aspect of decision-making is deciding whether it is worth initiating a voluntary action as opposed to doing nothing at all. While such decisions are ubiquitous in everyday human and animal life, and easily observed (although not always studied) in the animal laboratory, they are difficult to investigate in experiments with humans. Typically, social demand characteristics of the experimental setting make it unlikely that a participant will decide not to bother making an action that they have been asked to make even when they perform experiments in which they freely choose when to act. The present behavioural paradigm, however, led participants to make predictable decisions about when to act and when not to bother. We ensured this by allowing the participants to carry on viewing a movie when they decided to do nothing. First, we showed that participants took a variety of contextual factors into account when deciding whether to act and perform an effortful task for a potential reward. Willingness-to-act on each trial was defined as the probability of acting predicted

by the combination of opportunity and environment in a given context. The factors that we hypothesised might influence decisions to act were all found to exert clear effects on act/no act decisions; participants were more likely to give up watching the movie and exert effort for potential reward when they received high magnitude and high probability opportunities, and when the average value of offers in their current environment was low. Variation in the movie over time might have contributed to variation in act/no-act decisions. Despite the fact that any such variation remained unmodelled in our analyses (it was not correlated with task events and it remained as unaccounted noise in the GLMs), the factors that determined the value of acting were powerful enough for their impact to emerge clearly as statistically significant as explanations of both behaviour and of neural activity. Nevertheless, we conducted a control experiment to investigate the relationship between interest-in-the-movie-clip and willingness-to-act and showed that trial-by-trial variation in interest-in-the-movie-clip does not influence participants' decision about whether or not to act and therefore cannot confound interpretation of the data (Supplementary Fig. S1).

Next, after using ultra-high field imaging, we extracted BOLD signal from a group of a priori selected subcortical structures known to be involved in action planning and initiation including the dorsal striatum (DS), nucleus accumbens (NAc), midbrain dopaminergic system (MidD), pedunculopontine nucleus (PPN), habenula (HB), and the basal forebrain (BF). We found that BOLD activity in HB tracked the trial-by-trial variation in participants' willingness-to-act (Fig. 3). We could not find the same relationship in other structures.

HB is an ancient brain structure located at the caudal end of thalamus. It receives distributed inputs from the septum, basal ganglia, cingulate and insular cortex and sends direct and indirect projections to dopaminergic and serotonergic nuclei[22,23,28]. Functionally, HB is linked to avoidance of negative outcomes, omission of expected reward and control of impulsive behaviours[19]. Here we showed that the relationship between willingness-to-act and HB activity was strongest when participants received high value offers but nevertheless decided not to respond. This is consistent with a potential role of HB in impulse control and suppression of motor responses, accomplished by its inhibitory control of the dopaminergic midbrain[19,28,32].

Interestingly, our PPI analyses showed that participants' willingness-to-act influenced the functional relationship between HB and dopaminergic midbrain. However, while previous resting-state fMRI studies have reported more extensive HB functional connectivity with VTA than SNc[33,34], our PPI analysis showed that participants' willingness-to-act and their responses (act/no-act decisions) had more influence on the HB functional connectivity with SNc than VTA. This is consistent with previous findings regarding the role of SNc in self-paced action initiation[18] and the results from our previous 7 T study in which SN, but not VTA, was shown to encode initiation of self-paced actions[16]. HB is made up of a medial and a lateral subdivision with distinct connectivity profiles[28]. It is therefore important to note that while ultra-high field imaging enabled us to track activity in a small subcortical structure such as HB, there are still limits to its spatial resolution and thus our ability to distinguish between the medial and lateral subdivisions of HB. We therefore combined those subdivisions into a single HB region.

Anterior insula tracked participants' willingness-to-act on each trial—in parallel to HB. In addition, it encoded individual contextual factors in the environment that influenced willingness-to-act (Figs. 4, 5). Importantly, PPI analysis showed that participants' willingness-to-act modulated the functional connectivity between HB and the anterior insula. Together, these results

suggest that individual contextual factors in the environment were primarily tracked by the anterior insula, integrated, and then passed to HB to guide the decision about whether or not to act. Anterior insula is known to be involved in effort-based decision making and willingness to exert effort to obtain reward[35]. It is well placed to integrate various sources of contextual information; it receives diverse and multimodal sensory inputs, has reciprocal connections with limbic and frontal cortical structures implicated in valuation and, importantly, sends direct projections to HB[22,36]. In addition, whole-brain resting-state functional connectivity studies have reported tight functional coupling between HB and the salience network—a set of brain regions involved in detecting and integrating relevant internal and external stimuli—including the anterior insula[33]. In the context of voluntary action, however, in the current study it was possible to show that HB occupied a very specific position within a multi-layered network using structural equation modelling (SEM). Contextual information determining willingness-to-act is encoded in the anterior insula, converges on the habenula, and is then transmitted to SMA via the nigrostriatal pathway as the binary decision either to act or not act emerges (Fig. 6).

Understanding the role that HB plays in determining whether and how the reward environment might elicit the onset of voluntary behaviour may shed light on changes in behaviour seen in the clinic. Impairments in willingness to initiate an action can cause symptoms such as apathy which are prominent in several major neurological and psychiatric disorders including depression[10]. Interestingly, in patients with depression, the habenular activity was reported to be correlated with depression rating[37].

We have previously suggested that cholinergic basal forebrain (BF) regulates decisions about when to act by integrating current opportunities in the environment through its connections with the frontal cortex and the dopaminergic midbrain[16,17]. Here we propose that decisions about whether to act also involve a cortico-subcortical circuit, but now the critical subcortical component appears to be the HB rather than BF. HB receives direct projections from BF and is one of the few brain regions that controls both the dopaminergic and the serotonergic systems through direct and indirect projections. Therefore, it is ideally placed to receive contextual information from higher-level cortical areas and to control decisions about if and when to act through its interaction with the cholinergic, serotonergic and dopaminergic systems. Future studies should assess the possible distinctive role of these neuromodulatory systems in regulating one's willingness-to-act.

## Methods

**Subjects**. Twenty-five participants completed the study. They were aged 18−40 years, consisting of 7 males and 18 females. At the end of each testing session, they were paid £15 per hour for participating in the study. They could earn an additional £3−£7 depending on their performance during the task. All relevant ethical regulations for work with human participants were observed. At the beginning of each testing session, participants were required to provide a written informed consent. Ethical approval was given by the Oxford University Central University Research Ethics Committee (CUREC) (Ref-Number: MSD-IDREC-R55856/RE001). One subject was excluded from all behavioural and brain analyses for failing to respond frequently enough according to an a priori exclusion criterion (response rate of >85% or <15%). In addition, two subjects were excluded from all brain analyses due to excessive head motion (absolute mean displacement > 2 mm).

**Experimental task**. Before entering the scanner, participants received written instructions and were introduced to the task. This included a titration procedure to establish their maximum voluntary contraction (MVC), consisting in three trials on which participants exerted as much force as possible on a handheld dynamometer. Their MVC was defined as the maximal momentary force produced across the three attempts. When inside the scanner, participants watched a nature documentary—a randomly selected episode of Planet Earth—with audio provided

through MRI compatible headphones (Fig. 1a). Participants received a series of offers in the form of visual stimuli that were periodically superimposed upon the documentary. Upon accepting an offer, the movie was interrupted, and participants needed to exert a short bout of physical effort using the dynamometer in order to be eligible for their reward. If an offer was rejected, participants simply continued watching the documentary uninterrupted for an equivalent duration. Each offer appeared for 2 s and could be accepted by button-pad response while it remained on-screen. Offer stimuli consisted in centrally presented vertical rectangles which contained small circular dots (range = [1, 21]). The colour of the offer-stimulus represented reward-magnitude: either low (5p), medium (10p) or high (20p), as indicated by red, green or blue (magnitude-to-colour contingencies were counterbalanced across participants). The number of dots comprising the stimulus indicated the reward-probability, such that reward-probability increased linearly with the number of dots—for example, an offer with the maximum number of dots (21) was certain to yield reward. Both reward magnitude and reward probability varied from trial-to-trial in a pseudorandomised order (Fig. 1b). The task included a total of 216 offers (trials), which were divided into 6 blocks of 36 offers. The ratio of high magnitude and high probability offers within blocks was manipulated in order to influence the average value of the participants' environments—that is, the average value of the offers they could expect to receive in the near future. Rich blocks included higher ratios of high magnitude and high probability offers (50% high; 33% mid; 16% low), while the reverse was true in poor blocks (16% high; 33% mid; 50% low). Block type (rich vs poor) alternated sequentially and was counterbalanced across participants (Fig. 1b). Participants were informed at the beginning of each block as to whether they were entering a rich or a poor block.

If an offer was accepted, the film was interrupted, and the participant needed to complete a short effort-task. The effort-task involved squeezing the dynamometer at a target level of force corresponding to 50% of the participant's MVC (this ensured the effort level was always achievable but still physically demanding) for a short duration, randomly sampled from a Gaussian distribution (effort-time ~ $\mathcal{N}$(3.5 s, 0.5 s)). Real-time feedback about the level of force being produced relative to the target was provided via an on-screen force-metre which replaced the film (see Fig. 1a). Force data were recorded using a TSD121B-MRI dynamometer (BIOPAC Systems Inc., USA) running on an MP160 acquisition device (BIOPAC Systems Inc., USA). If the force was successfully exerted, participants became eligible for a monetary reward with the probability and at the magnitude indicated by the accepted offer. If it was not, the participant was ineligible for a reward and was informed of their failure to exert the requisite force. The movie resumed once the effort-task was completed, and feedback about the reward outcome was superimposed on-screen for 2 s. If the offer was not accepted, participants continued to watch the movie uninterrupted for a duration equivalent to the effort-task but would receive no monetary reward. Feedback about the reward outcome (0p) was superimposed on the screen for 2 s, even when participants did not accept the offer. The next offer (trial) appeared after an inter-trial-interval (ITI) randomly sampled from a Gaussian distribution (ITI ~ $\mathcal{N}$(4.5 s, 0.5 s)). The task finished after six blocks (three poor blocks + three rich blocks). The experiment was written in Matlab (Mathworks, Natick, USA), using the Psychophysics Toolbox extension[38].

**Behavioural analysis**. Willingness-to-act on each trial was defined as the probability of acting, by making a response to accept an offer, given the combination of contextual factors. The contextual factors included reward magnitude, probability, block type (rich/poor) on the current trial, and the response (act/no-act) and expected value (reward magnitude × probability) on the immediately preceding trial. We used a generalised linear mixed-effect model to estimate the regression coefficients. The modelling was performed with the lme4 and optimx packages in R[39,40].

$$\begin{aligned}\text{logit}(\text{response}_i) = &\ \beta_0 + \beta_1 \text{reward}_i + \beta_2 \text{probability}_i + \beta_3 \text{block}_i + \beta_4 (\text{reward}_i \times \text{probability}_i) \\ &+ \beta_5 (\text{reward}_i \times \text{block}_i) + \beta_6 (\text{block}_i \times \text{probability}_i) + \beta_7 \text{expectedValue}_{i-1} + \beta_8 \text{response}_{i-1} \\ &+ \beta_9 \text{totalTime}_i + \mu_0 + \mu_1 \text{reward}_i + \mu_2 \text{probability}_i + \mu_3 \text{block}_i + \mu_4 (\text{reward}_i \times \text{probability}_i) \\ &+ \mu_5 (\text{reward}_i \times \text{block}_i) + \mu_6 (\text{block}_i \times \text{probability}_i) + \mu_7 \text{expectedValue}_{i-1} + \mu_8 \text{response}_{i-1} \\ &+ \mu_9 \text{totalTime} + e, \end{aligned}$$

$$(1)$$

where $\beta_{0-9}$ are the fixed effects, $\mu_0$ is by-subject random intercept, $\mu_{1-9}$ are by-subject random slopes, and $i$ is the trial number. reward is the offered reward-magnitude (low, medium or high); probability is the reward probability as indicated by number of dots; block is the average value of the participants' environments (rich or poor); expected value is the product of reward and probability; response is the act/no-act decision. Total time (from beginning of the testing session up to the current trial, totalTime) was added to the model as a covariate of no interest to account for fatigue. logit is defined as:

$$\log\left(\frac{p(X)}{1 - p(X)}\right) \tag{2}$$

where $X = (X_1, \ldots, X_p)$ are $p$ predictors.

To measure willingness-to-act, we first fit the predictors of the above model separately to each participant's response.

$$\text{logit}(\text{response}_{s,i}) = \beta_0 + \beta_1 \text{reward}_{s,i} + \beta_2 \text{probability}_{s,i} + \beta_3 \text{block}_{s,i} + \beta_4(\text{reward}_{s,i} \times \text{probability}_{s,i})$$
$$+ \beta_5(\text{reward}_{s,i} \times \text{block}_{s,i}) + \beta_6(\text{block}_{s,i} \times \text{probability}_{s,i}) + \beta_7 \text{expectedValue}_{s,i-1}$$
$$+ \beta_8 \text{response}_{s,i-1} + \beta_9 \text{totalTime}_{s,i} + e,$$

$$(3)$$

where $s$ is the subject number. Next, we used the estimated coefficients ($\beta_{0-9}$) to measure the willingness-to-act separately for each participant and each trial by using a logistic function:

$$\text{willingness to act}_{s,i} = p(\text{response}_{s,i}) = e^{\beta_0 + \beta_1 \text{reward}_{s,i} + \cdots + \beta_9 \text{totalTime}_{s,i}} / 1 + e^{\beta_0 + \beta_1 \text{reward}_{s,i} + \cdots + \beta_9 \text{totalTime}_{s,i}}$$

$$(4)$$

**Imaging data acquisition**. Structural and functional MRI was collected using a Siemens 7 T MRI scanner. High-resolution functional data were acquired using a multiband gradient-echo T2* echo planar imaging (EPI) sequence with a 1.5 × 1.5 × 1.5 mm resolution; multiband acceleration factor 3; repetition time (TR) 1962 ms; echo time (TE) 20 ms; flip angle 66°; and a GRAPPA acceleration factor 2. Field of view (FOV) was adjusted to cover the whole-brain with axial orientation and a fixed angulation of −30° (anterior-to-posterior phase encoding direction; 96 slices). Additionally, a single-measurement, whole-brain, functional image was acquired prior to the main functional image (with similar orientation). This pre-saturation scan was later used to improve registration of the main functional image to the whole brain. Structural data were acquired with a T1-weighted MP-RAGE sequence with a 0.7 × 0.7 × 0.7 mm resolution; GRAPPA acceleration factor 2; TR 2200 ms; TE 3.02 ms; and inversion time (TI) 1050 ms. To correct for field inhomogeneities a separate Fieldmap sequence was acquired with a 2 × 2 × 2 mm resolution; TR 620 ms; TE1 4.08 ms; TE2 5.10 ms. Finally, to regress out the effect of physiological noise in functional data, cardiac and respiratory frequencies were collected by pulse oximetry and respiratory bellows.

**fMRI data processing**. Pre-processing was performed using tools from FMRIB Software Library (FSL)[41]. Functional images were first normalised, spatially smoothed (Gaussian kernel with 3 mm full-width half-maximum), and temporally high-pass filtered (3 dB cut-off of 100 s). The effect of participants' head motion during the scanning was removed by MCFLIRT[42]. The Brain Extraction Tool (BET)[43] was used on functional and structural images to separate brain from non-brain matter. The registration of functional images into Montreal Neurological Institute (MNI)-space was performed in three stages: (1) whole-brain task EPI to pre-saturation EPI using FMRIB's Linear Image Registration Tool[44] with three degrees of freedom (translation only). (2) Whole-brain EPI to individual structural image using Boundary-Based Registration (BBR)[45] by incorporating Fieldmap correction. (3) Individual structural image to Standard image by using FMRIB's Non-linear Image Registration Tool (FNIRT).

**Whole-brain fMRI data analyses**. Whole-brain statistical analyses were performed at two-levels as implemented in FSL FEAT[46]. At the first level, we used a univariate general linear model (GLM) framework for each participant to compute the parameter estimates. The contrast of parameter estimates and variance estimates from each scanning session were then combined in a second-level mixed-effects analysis (FLAME 1 + 2)[47], treating scanning sessions as random effect. The results were cluster-corrected with the voxel inclusion threshold of $Z = 3.1$ and cluster significance threshold of $P = 0.001$. The data were pre-whitened before analysis to account for temporal autocorrelations[46].

The first-level analyses looked for voxels, across the whole brain, in which BOLD signal reflected parametric variation in willingness-to-act (i.e., the probability of acting given the combination of contextual factors).

$$\text{GLM1} : \text{BOLD} = \beta_0 + \beta_1 \text{stim} + \beta_2 \text{willingness\_to\_act} + \beta_3 \text{response} + \beta_4 \text{effort} + \beta_5 \text{mainOut} + \beta_6 \text{reward}$$
$$+ \beta_7 \text{nonConvResp} + \beta_8 \text{nonConvOut},$$

$$(5)$$

where BOLD is a $t \times 1$ ($t$ time samples) column vector containing the times-series data for a given voxel. stim is an unmodulated regressor representing the main effect of stimulus (offer) presentation (all event amplitudes set to one). willingness_to_act is a parametric regressor representing willingness-to-act (see "Behavioural analysis"). response is a binary regressor with two levels (act/no-act). effort is a parametric regressor representing the amount of force participant exerted during the effort-task. mainOut is an unmodulated regressor representing the main effect of outcome. reward is a binary regressor with two levels (rewarded/not rewarded) representing the reward outcome on the current trial. Regressors 1−6 were modelled as a boxcar function with constant duration of 0.1 s convoluted with a double-gamma hemodynamic response function (HRF). Regressors 1−2 were time-locked to the onset of the trial (presentation of the offer). Regressor 3 was time-locked to the moment the participant made a response, if the offer was accepted, and to 2 s after the trial onset, if the offer was rejected. Regressor 4 was time-locked to the onset of the effort-task in trials where the offer was accepted. Regressors 5−6 were time-locked to the onset of the reward outcome phase. To

model instant signal distortions due to changes in the magnetic field caused by performing the effort-task we added two additional constant regressors that were not convoluted with HRF (nonConv). These regressors started at the beginning of the TR when the response was recorded and the effort-task started (nonConvResp), and when the outcome phase started (nonConvOut). They had a duration of one TR (1.96 s).

To further reduce variance and noise in the BOLD signal, we also added task-unrelated confounds which included: (1) head motion parameters as estimated by MCFLIRT in the pre-processing stage; (2) voxelwise regressors created by physiological noise modelling (PNM)[48] to model the effects of physiological noise (cardiac and respiratory); (3) regressors to completely remove timepoints with large motions that could not be fixed with linear methods (across participants, 5 ± 3% of timepoints were marked as corrupted by large motion).

**ROI time-course analyses**. To create anatomical regions of interest (ROI) anatomical masks were designed for each ROI in the MNI standard space using the Harvard-Oxford Subcortical Structural Atlas and Atlas of the Human Brain[49] (Fig. 2). Masks were then transformed from the standard space to each participant's structural space by applying a standard-to-structural warp field and from structural to functional space by applying a structural-to-functional affine matrix. Transformed masks were thresholded, binarised and were dilated by one voxel. Functional ROIs (anterior insula and SMA) were defined as spheres of 1.5 mm radius, centred at the centroid of local maxima (peaks) of an activation cluster.

For time-series analysis the following steps were followed: (1) the filtered time-series from each voxel within each ROI was extracted; (2) the data were then averaged across the voxels, normalised and up-sampled 20 times; (3) the up-sampled data were interpolated using the cubic spline method; (4) the interpolated data were then epoched in 10 s windows, starting from 2 s before to 8 s after the trial onset (appearance of the offer); (5) finally, ordinary least squares (OLS) method was used to fit the GLMs at each time step of the epoched data. We ran the following GLMs:

$$\text{GLM2.1} : \text{BOLD} = \beta_1 \text{willingness\_to\_act} + \beta_2 \text{response} + \beta_3 \text{constant}, \quad (6)$$

where BOLD is a $i \times t$ ($i$ trial, $t$ time samples) matrix containing the times-series data for a given ROI. willingness_to_act is the probability of acting for a given opportunity in a given environment. response is a dummy-coded variable representing the participants' observed response (act/no-act decision). constant is an unmodulated constant regressor.

$$\text{GLM2.2} : \text{BOLD} = \beta_1 \text{reward}_i + \beta_2 \text{probability}_i + \beta_3 \text{block}_i + \beta_4 \text{expectedValue}_{i-1} + \beta_5 \text{response}_{i-1}$$
$$+ \beta_6 \text{totalTime}_i + \beta_7 \text{ response}_i + \beta_8 \text{ constant},$$

$$(7)$$

where $\text{reward}_i$, $\text{probability}_i$ and $\text{block}_i$ are reward magnitude, reward probability and block type (average value of the environment), respectively, at trial $i$. $\text{expectedValue}_{i-1}$ and $\text{reponse}_{i-1}$ are expected value and response (act/no-act decision) at the preceding trial ($i-1$). $\text{totalTime}_i$ is the total time passed from beginning of the testing session up to the current trial. $\text{response}_i$ is the observed act/no-act decision at trial $i$.

$$\text{GLM2.3} : \text{BOLD}_{\text{ROI}} = \beta_1 \text{BOLD}_{\text{seed}} + \beta_2 \text{willingness\_to\_act} + \beta_3 \text{PPI} + \beta_4 \text{ response} + \beta_5 \text{ constant},$$

$$(8)$$

where $\text{BOLD}_{\text{ROI}}$ is BOLD activity at anterior insula, $\text{BOLD}_{\text{seed}}$ is BOLD activity at HB, and PPI is the interaction between $\text{BOLD}_{\text{seed}}$ and willingness_to_act.

$$\text{GLM2.4} : \text{BOLD} = \beta_1 \text{BOLD}_{\text{seed}} + \beta_2 \text{willingness\_to\_act} + \beta_3 \text{response} + \beta_4 \text{PPI} + \beta_5 \text{constant}, \quad (9)$$

where $\text{BOLD}_{\text{ROI}}$ is BOLD activity at MidD, $\text{BOLD}_{\text{seed}}$ is BOLD activity at HB, and PPI is a three-way interaction between $\text{BOLD}_{\text{seed}}$, willingness_to_act, and response.

**Leave-one-out analysis on time-series group peak signal**. Significance testing on time-course data was performed by using a leave-one-out procedure on the group peak signal to avoid potential temporal selection biases. For every participant, we estimated the peak signal time by identifying the peak in the time course of the mean beta weights of the relevant regressor in all other participants. When we did this, we identified the peak (positive or negative) of the regressor of interest within the full width of the epoched time course: from 2 s before to 8 s after the trial onset. Next, we took the beta weight of the remaining participant at the time of the group peak. We repeated this for all participants. Therefore, the resulting 22 peak beta weights were selected independently from the time course of each single participant. We assessed significance using $t$ tests on the resulting peak beta weights. All $t$ tests were two-sided. To control for familywise error rate the significance level was adjusted, whenever doing three or more comparisons, using the Holm−Bonferroni method[50]. This method was the only type of correction that was applied to all the analyses—whenever doing three or more comparisons—regardless of the GLM and ROI: e.g., when comparing across six ROIs as in Figs. 2 and 3 or across four contextual features as in Fig. 5.

**Structural equation modelling**. PPI analysis is applicable to a maximum of two ROIs at a time[30]. To investigate how regions are connected at a wider circuit level, we conducted structural equation modelling (SEM) to probe for covariance

between regions in the time-course of BOLD response. SEM assesses inter-relationships among several continuous variables based on their covariance with one another. Importantly, it defines the strength of connections between brain areas in question, rather than activity in individual variables. In practice, it is reminiscent of linear regression save that more complex patterns of relationships can be tested. As implemented here, it has the following formal definition:

$$\eta_t = \eta_t \cdot \beta + \zeta_t, \qquad (10)$$

where $\eta_t$ is a matrix comprising the filtered time-series of BOLD response in the ROIs, $\beta$ is a vector of path coefficients describing the relationship(s) between ROIs, and $\zeta_t$ is measurement error. All structural equation modelling was conducted in Latent Variable Analysis (lavaan) package v.0.6-4 using Maximum Likelihood estimation[51]. Akaike information criterion (AIC) was used for model comparison.

**Reporting summary**. Further information on research design is available in the Nature Research Reporting Summary linked to this article.

## Data availability

Data files and materials used in the main analyses presented here (Figs. 1c–f, 2, 3, 4b–d, 5) have been archived and uploaded to the Data DRYAD and are freely available at: https://doi.org/10.5061/dryad.6t1g1jwxq (ref. [52]). Source data are provided with this paper.

## Code availability

Custom-written R scripts used for measuring willingness-to-act and reproducing the figures related to behavioural analysis are available at https://doi.org/10.5061/dryad.6t1g1jwxq.

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

## Acknowledgements
This study is funded by Wellcome Trust grants WT101092MA, WT100973AIA, 203139/Z/16/Z. N.G. is funded by a Sir Henry Wellcome Postdoctoral Fellowship.

## Author contributions
N.K., N.G. and M.F.S.R. conceptualised the study; L.P. implemented the experiment; N.K. and L.P. collected and analysed the data; N.K., L.P., N.G. and M.F.S.R. interpreted the data; P.L. provided study materials; N.K. drafted the paper; all authors revised and approved the final version of the paper.

## Competing interests
The authors declare no competing interests.
