## [Peer Review File · Nature Communications]

A habenula-insular circuit encodes the willingness to actReviewers' comments:

Reviewer #1 (Remarks to the Author):

Khalighinejad and colleagues have submitted a fascinating manuscript entitled 'A habenula-insular circuit encodes whether it is worth acting.' The study reports significant new results from a 7T fMRI neuroimaging experiment. The task is simple, but the experiments are cleverly designed and performed to a high technical standard that is described in sufficient detail. The manuscript is well-written and clear, and the authors cite and discuss the relevant literature in a balanced way. The key novel finding is that activity in the habenula tracked the participants' willingness to act. The habenula is rarely studied in human neuroimaging studies, and thus the present study advances our understanding of the functional role of this small brain region. Moreover, by using effective connectivity analyses, the authors can show how the habenula works in concert with the insular cortex and the supplementary motor area to generate the decisions that shape the willingness to act. In sum, this is an excellent neuroimaging study.

Reviewer #2 (Remarks to the Author):

The study investigates an interesting question related to the neural correlates of intentional action, namely which brain regions are involved in the willingness-to-act when people can freely decide whether to execute an effortful behavior or not. In a region-of-interest analysis of subcortical structures, it is found that willingness-to-act is significantly related to the habenula. A whole brain analysis indicates that willingness-to-act is also related to the anterior insula and SMA. A PPI analysis revealed a modulation of the relationship between habenula and anterior insula by willingness-to-act but not between habenula and SMA. Finally, structural equation modelling was used to test different multi-layer networks that best explain the data. While this is an interesting study using a clever experimental design, I have some major concerns regarding the interpretation of the results.

The prominent role of the habenula in tracking willingness-to-act is inferred from the fact that this was the only subcortical region that showed a significant relationship with willingness-to-act. However, in order to really draw strong conclusions about the specific role of the habenula one should either show that the other regions do not show such a relationship (showing a convincing null effect in these regions) or at least that the habenula shows a stronger relationship than the other regions. But this is not what the authors demonstrate.

The authors found a stronger relationship between habenula and willingness-to-act for trials where participants refrained from acting compared to trials where they acted. The authors inferred from this a potential role of the HB in impulse control and suppression of action. In my opinion, this conclusion is not justified. The stronger relationship could be simply due to the fact that the variation of willingness-to-act was different for trials in which participants acted compared to trials in which they did not act. In my opinion, the stronger relationship in trials in which participants did not act does not justify a conclusion regarding the involvement of the habenula in refraining from acting.

Along this line, I think that the conclusion 'that the relationship between willingness-to-act and HB activity was strongest when participants received high value offers but nevertheless decided not to respond' is not evident (at least if it based on the stronger relationship of HB activation and willingness to act in trials where they did not act).

I am not an expert in SEM but I wonder whether the difference in AIC between Model 1 and 3 is large enough to favor Model 1.

Reviewer #3 (Remarks to the Author):

This is an interesting study on deep brain activation patterns during decision-making. The study,

however, exhibits two major flaws. First, willingness-to-act is defined as the probability of acting, by making a response to accept an offer, given the combination of contextual factors. While these contextual factors include reward magnitude and probability, they the most obvious factor: interest in the movie. Obviously, the authors need to differentiate between movie scenes with little/no and high interest to the participants. This level of interest will not only vary across participants, but also intra-individually, depending on the trial-specific movie content. Without this information, the study is incomplete and the current interpretation of fMRI findings flawed. The second and more severe flaw relates to negative activation found in the dopaminergic midbrain (MidD). While the authors acknowledge that negative MidD is both counterintuitive and against previous reports, the authors fail to investigate this issue thoroughly. Instead, they state that "some suggest that a negative signal is a result of neuronal inhibition others suggest that it might actually be a sign of increased neuronal activity". The authors neglect the possibility that the same interpretation could be stated to their other results. Even more, these considerations are hidden in the caption of a supplementary figure. The authors need to show clear data on the baseline issues the claim as explanation for this negative activation patterns, particularly why this should affect MidD, but not DS and HB.

Specific issues:

Intro: "Behavioural analysis showed that opportunities in a given environment influenced participants' willingness to break away from the movie and undertake effort in return for potential reward". This statement is confusing as neither opportunities nor willingness are defined sufficiently.

Intro: "BOLD activity in HB tracked participants' willingness-to-act". Authors should avoid using "Tracked" as this implies a temporal sequence. Please use "correlates" instead.

Results: "Participants (N=25) watched a movie". This statement is misleading. According to the method's section, 3 subjects were excluded from fMRI analyses (1 for inadequate response patterns, 2 due to excessive motion). While it is true that 25 subjects participated in the study, fMRI results are based on 22 subjects. When presenting results, authors must explicitly state the number of subjects their results are based on. Providing this information in the methods' section is not sufficient.

Methods, linear mixed-effects models. Index "t" should not be used when referring to "trial" as most readers will associate "t" with "time". Please use "i" or "j" instead.

Methods, linear mixed-effects models. $\text{logit}(\text{response})$ is confusing. Please add the detailed equation.

Methods, linear mixed-effects models. Line 432. Please provide detailed information on reward_t , probability_t , block_t etc.

Methods, line 408, "from a Gaussian distribution (effort-time $\sim \mathcal{N}(3.5, 0.5)$)." Please add units (i.e. seconds) to the distribution description.

Methods, line 529. Please provide details on the "up-sampling" process.

Methods, multiple comparison correction. Please provide clear and detailed information on the multiple comparison corrections applied. The authors tested several GLMs across several ROIs. In the current description, the particular types of corrections applied is not clear.

Figure 1: integrating over the histogram of mean acceptance rates yields 24 subjects (not 25)

Reviewer #1:

Khalighinejad and colleagues have submitted a fascinating manuscript entitled 'A habenula-insular circuit encodes whether it is worth acting.' The study reports significant new results from a 7T fMRI neuroimaging experiment. The task is simple, but the experiments are cleverly designed and performed to a high technical standard that is described in sufficient detail. The manuscript is well-written and clear, and the authors cite and discuss the relevant literature in a balanced way. The key novel finding is that activity in the habenula tracked the participants' willingness to act. The habenula is rarely studied in human neuroimaging studies, and thus the present study advances our understanding of the functional role of this small brain region. Moreover, by using effective connectivity analyses, the authors can show how the habenula works in concert with the insular cortex and the supplementary motor area to generate the decisions that shape the willingness to act. In sum, this is an excellent neuroimaging study.

Reviewer #2:

The study investigates an interesting question related to the neural correlates of intentional action, namely which brain regions are involved in the willingness-to-act when people can freely decide whether to execute an effortful behavior or not. In a region-of-interest analysis of subcortical structures, it is found that willingness-to-act is significantly related to the habenula. A whole brain analysis indicates that willingness-to-act is also related to the anterior insula and SMA. A PPI analysis revealed a modulation of the relationship between habenula and anterior insula by willingness-to-act but not between habenula and SMA. Finally, structural equation modelling was used to test different multi-layer networks that best explain the data. While this is an interesting study using a clever experimental design, I have some major concerns regarding the interpretation of the results.

1. The prominent role of the habenula in tracking willingness-to-act is inferred from the fact that this was the only subcortical region that showed a significant relationship with willingness-to-act. However, in order to really draw strong conclusions about the specific role of the habenula one should either show that the other regions do not show such a relationship (showing a convincing null effect in

these regions) or at least that the habenula shows a stronger relationship than the other regions. But this is not what the authors demonstrate.

The habenula was indeed the only area that showed significant relationship with willingness-to-act ($t(21)=4.26$, $P=0.002$). Other regions did not show such a relationship: DS, $t(21)=1.46$, $P=0.48$; NAc, $t(21)=0.05$, $P=0.96$; MidD, $t(21)=0.94$, $P=0.72$; PPN, $t(21)=1.93$, $P=0.27$; BF, $t(21)=2.41$, $P=0.13$ (corrected for multiple comparisons). Thus, exactly the results requested had been reported in the manuscript. However, we have now revised the paragraph and Fig.3 to put more emphasis on the null results (line 153):

We found that trial-by-trial variation in participants' willingness-to-act explained BOLD activity in HB ($t(21)=4.26$, $P=0.002$, $d=0.90$; corrected for multiple comparisons) (Fig.3; also see Fig.S2 for alternative HB mask). **This relationship was not found in any other ROI (all $P>0.12$; also see Table S2 for Bayesian analysis).**

Additionally, we used Bayesian statistics to re-investigate the null effects. As shown in the table below there is 'very strong' evidence in support of our hypothesis in HB ($BF_{10}>30$) but 'no evidence' in other areas ($BF_{10}<1$). There is 'anecdotal evidence' in basal forebrain (BF), but it is evident in figure 3 that the positive deflection in BF happens before stimulus presentation and therefore could not be related to encoding of willingness-to-act. We have added these results to the revised manuscript too (see table S2).

Bayesian One Sample T-Test

ROI	BF_{10}	error %
DS	0.562	0.019
NAc	0.223	0.031
MidD	0.330	0.030
PPN	1.064	0.005
HB	91.238	6.304e-5
BF	2.312	4.602e-4

Note. For all tests, the alternative hypothesis specifies that the population mean differs from 0.

2. The authors found a stronger relationship between habenula and willingness-to-act for trials where participants refrained from acting compared to trials where they acted. The authors inferred from this a potential role of the HB in impulse control and suppression of action. In my opinion, this conclusion is not justified. The stronger relationship could be simply due to the fact that the variation of willingness-to-act was different for trials in which participants acted compared to trials in which they did

not act. In my opinion, the stronger relationship in trials in which participants did not act does not justify a conclusion regarding the involvement of the habenula in refraining from acting. Along this line, I think that the conclusion ‘that the relationship between willingness-to-act and HB activity was strongest when participants received high value offers but nevertheless decided not to respond’ is not evident (at least if it based on the stronger relationship of HB activation and willingness to act in trials where they did not act).

The alternative explanation suggested by the reviewer is plausible but easy to investigate. We compared the variation of willingness-to-act between the act and not-act trials. We found no evidence in support of the reviewers’ hypothesis: the mean variation across participants in willingness-to-act for action trials was 0.23 (± 0.05). It was 0.21 (± 0.04) for trials in which participants did not act. Importantly, the difference between the two was not statistically significant: paired-samples t-test; $p=0.12$.

This analysis suggests that the stronger relationship between HB and willingness-to-act could not be simply explained by the difference in variation in willingness-to-act between act and no-act trials.

The following paragraph was added to the legend of Fig.S3:

Figure S3. Related to Figure 3. The effect of willingness-to-act on HB BOLD signal plotted separately for trials in which participants decided to make an action (response), and those in which they withheld an action (refrained from responding). **The stronger relationship between HB activity and willingness-to-act in non-responded trials could be simply due to the fact that the variation of willingness-to-act was different for trials in which participants acted compared to trials in which they did not act. To test this hypothesis, we compared the variation of willingness-to-act between the act and not-act trials. The mean variation across participants in willingness-to-act for action trials was 0.23 (± 0.05). It was 0.21 (± 0.04) for trials in which participants did not act. Importantly, the difference between the two was not statistically significant: paired-samples t-test; $p=0.12$. This suggests that the stronger relationship between HB and willingness-to-act in non-responded trials could not be simply explained by the difference in variation in willingness-to-act.**

The full set of results pertaining to the habenula (HB) and substantia nigra par compacta (SNc) are summarized in the following new addition to the manuscript, line 264:

In summary, the results discussed here and in Figures 3 and S1 reveal that HB encodes willingness-to-act at the time that an action cue first appears although SNc does not. However, activity coupling begins to occur between HB and SNc as a function of willingness to act, and then an activity increase occurs in SNc that is time-locked to the moment of movement onset.

3. I am not an expert in SEM but I wonder whether the difference in AIC between Model 1 and 3 is large enough to favor Model 1.

We agree with the reviewer that difference between model 1 and 3 is less substantial than the difference between model 1 with 2&4. However, the difference in AIC between model 3 and 1 (ΔAIC) remains very substantial: $377651 - 376817 = 834$. $\Delta AIC > 10$ indicates essentially no support for the model with the larger AIC, and a great deal of support for the model with the smaller AIC (Burnham & Anderson, 2002).

Reviewer #3:

This is an interesting study on deep brain activation patterns during decision-making. The study, however, exhibits two major flaws.

1. First, willingness-to-act is defined as the probability of acting, by making a response to accept an offer, given the combination of contextual factors. While these contextual factors include reward magnitude and probability, they the most obvious factor: interest in the movie. Obviously, the authors need to differentiate between movie scenes with little/no and high interest to the participants. This level of interest will not only vary across participants, but also intra-individually, depending on the trial-specific movie content. Without this information, the study is incomplete and the current interpretation of fMRI findings flawed.

It is true that the trial-specific movie scenes could have influenced decisions to act or not to act across participants and also intra-individually. However, we have taken great care in the inferences that we have drawn in the study and the inferences we have made remain secure despite this concern.

The movie shown to each participant was a ‘randomly’ selected episode of a nature documentary. Importantly, the parametric design and the mixed-effect modelling of the behaviour and brain (which included both by-subject random intercepts and random slopes), strongly controlled for any across-participant variance in interest in the movie. As for intrasubject variance it is important to emphasise that there was no relationship between movie scenes and the task parameters and therefore interest in movie scenes could not have any ‘systematic’ influence on participants’ decisions. Variation in the movie did of course contribute to the ‘unexplained variance’ in our predictive models. Critically, however, the experimentally controlled factors that determined the value of acting (i.e., reward, probability, block type) were powerful enough for their impact to emerge clearly as statistically significant as explanations of both behaviour and of neural activity. This means that while interest in movie scenes adds to the proportion of ‘unexplained variance’ in the behaviour and brain data, this does not add a flaw to the interpretation of that proportion of variance in the behaviour and brain that ‘could be explained’ by our experimentally controlled factors.

We have now added two paragraphs in the Results and Discussion to explain this point.

Results line 80:

In summary, we experimentally manipulated three factors in order to alter participants’ environments and influence their decision: reward magnitude, reward probability and block type. Our focus was on establishing the factors that lead people to act and so we carefully quantified those factors that would lead them to act such as the potential reward benefits, probabilities and the costs. Importantly, at the same time there was no relationship between movie scenes and the task parameters and therefore interest in movie scenes could not have any systematic influence on participants’ decisions.

Discussion line 318:

The factors that we hypothesized might influence decisions to act were all found to exert clear effects on act/no act decisions; participants were more likely to give up watching the movie and exert effort for potential reward when they received high magnitude and high probability opportunities, and when the average value of offers in their current environment was low. Variation in the movie over time might have contributed to variation in act/no-act decisions. Despite the fact that any such variation remained unmodelled in our analyses (it was not correlated with task events and it remained as unaccounted noise in the GLMs), the

factors that determined the value of acting were powerful enough for their impact to emerge clearly as statistically significant as explanations of both behaviour and of neural activity.

It is possible that a future study might systematically vary the interest of each movie scene shown to participants. Such ratings might be used to index a variable that could be labelled “unwillingness-to-act” and which might be conceptualized as a potential complement to the factor of “willingness-to-act” that we have studied here. However, whether such an experimental manipulation would yield ecologically meaningful insight into the slowly varying motivational processes that normally make a person disinclined to act is not clear.

2. The second and more severe flaw relates to negative activation found in the dopaminergic midbrain (MidD). While the authors acknowledge that negative MidD is both counterintuitive and against previous reports, the authors fail to investigate this issue thoroughly. Instead, they state that “some suggest that a negative signal is a result of neuronal inhibition others suggest that it might actually be a sign of increased neuronal activity”. The authors neglect the possibility that the same interpretation could be stated to their other results. Even more, these considerations are hidden in the caption of a supplementary figure. The authors need to show clear data on the baseline issues they claim as explanation for this negative activation patterns, particularly why this should affect MidD, but not DS and HB.

We agree that the negative activation in MidD in regard to action initiation is confusing and warrants further investigations. One important point that was overlooked in the original version of the paper was that we were considering activity that was time-locked to the stimulus onset rather than movement onset. It is, however, at the time of movement onset when MidD activity is most clearly expected and it is, arguably, particularly within one subsection of MidD – the substantia nigra pars compacta – that it is expected. When, by contrast, the analysis focused on activity time-locked to movement onset and carefully examined activity in the two subsections of MidD – the substantia nigra pars compacta (SNc) and the ventral tegmental area (VTA) (see figure below) – we found clear evidence of activity related to movement onset in SNc as might have been expected from previous reports. Therefore, the negative activation found in MidD – that is time-locked at stimulus onset and which is seen when averaging activity across MidD – is not indicative of a major flaw in the study: the SNc division of MidD encodes action initiation, just as the reviewer envisaged, but it does so at the time of action.

We added this new finding to the Results line 140:

“Here, however, we found a negative relationship between act/no-act decisions and BOLD activity in MidD (see Fig.2a). This might at first seem counterintuitive given that one would expect MidD to positively encode the ‘act/no-act’ decisions. The key point to note, however, is that we are considering activity that is time-locked to the stimulus onset rather than movement onset. When, by contrast, we focus on activity time-locked to movement onset and carefully examine activity in the two subsections of MidD – the substantia nigra pars compacta (SNc) and the ventral tegmental area (VTA) – we see clear evidence of activity related to movement onset in SNc as might have been expected given that many researchers²³, including ourselves¹⁶, have previously identified SNc with action initiation (Fig.S1).”

We also replaced Fig.S1 with a new figure:

Figure S1. Related to Figure 2. We found a negative relationship between act/no-act decisions and BOLD activity in MidD (see Fig.2a). This might be counterintuitive given that others²³ and our own findings¹⁶ have previously showed that MidD encodes action initiation. Accordingly, one would expect MidD to positively encode the ‘response’ effect. One way to address this discrepancy directly is simply to separate the subdivisions of MidD (substantia nigra pars compacta (SNc) and ventral tegmental area (VTA) based on the Pauli Atlas²⁴; see Methods) and investigate the effect of ‘response’ (act/no-act decisions) on BOLD signals

time locked to action onset (rather than stimulus onset as in Figure 2). We found – in accordance with previous reports – a positive deflection in SNc when time-locking to action onset (a,b) (GLM2.1). The positive SNc activity – when time locked to action onset – corresponds with the role of SNc in action initiation, once the haemodynamic lag is taken into account. This shows that from the negative activity pattern that is time-locked at stimulus onset and which is seen when averaging activity across MidD (Fig.2), one cannot conclude that nowhere in MidD encodes action initiation; the SNc division of MidD encodes action initiation but it does so at the time of action. The lines and shadings show the mean and standard error of the β weights across the participants, respectively. Time zero is the action onset. Significance testing on time-course data was performed by using a leave-one-out procedure on the group peak signal. One-sample t-tests. * $P < 0.05$.

Specific issues:

3. Intro: “Behavioural analysis showed that opportunities in a given environment influenced participants’ willingness to break away from the movie and undertake effort in return for potential reward”. This statement is confusing as neither opportunities nor willingness are defined sufficiently.

This sentence is now revised. See line 51:

Behavioural analysis showed that cost and benefits of reward opportunities in a given environment influenced participants’ willingness to act and undertake effort in return for potential reward.

4. Intro: “BOLD activity in HB tracked participants’ willingness-to-act”. Authors should avoid using “Tracked” as this implies a temporal sequence. Please use “correlates” instead.

‘tracked’ was replaced with ‘correlated’. See line 54:

Ultra-high field functional imaging showed that BOLD activity in HB was correlated with participants’ willingness-to-act.

5. Results: “Participants (N=25) watched a movie”. This statement is misleading. According to the method’s section, 3 subjects were excluded from fMRI analyses (1 for inadequate response patterns, 2 due to excessive motion). While it is true that 25 subjects participated in the study, fMRI results are based on 22 subjects. When presenting results, authors must explicitly state the number of subjects their results are based on. Providing this information in the methods’ section is not sufficient.

This information remains clearly available in the Methods line 399 but it is now also added to the Results line 62:

Participants (N=25) watched a movie whilst inside a 7T scanner (one was later excluded from the behavioural and brain analysis and two from brain analysis; see Methods).

6. Methods, linear mixed-effects models. Index “t” should not be used when referring to “trial” as most readers will associate “t” with “time”. Please use “i” or “j” instead.

‘t’ was replaced with ‘i’

7. Methods, linear mixed-effects models. $\text{logit}(\text{response})$ is confusing. Please add the detailed equation.

The definition of logit was added to the Methods line 472.

8. Methods, linear mixed-effects models. Line 432. Please provide detailed information on reward_t, probability_t, block_t etc.

Detailed information was added. Please see Methods line 467.

9. Methods, line 408, “from a Gaussian distribution (effort-time $\sim (3.5, 0.5)$).” Please add units (i.e. seconds) to the distribution description.

The units were added.

10. Methods, line 529. Please provide details on the “up-sampling” process.

Additional information was added to the Methods line 562:

For time-series analyses, the filtered time-series of each voxel within each ROI was averaged, normalised and up-sampled (20 times). The up-sampled data was then interpolated using the cubic spline method and was epoched in 10 s windows, starting from 2 s before to 8 s after the trial onset (appearance of the offer).

11. Methods, multiple comparison correction. Please provide clear and detailed information on the multiple comparison corrections applied. The authors tested several GLMs across several ROIs. In the current description, the particular types of corrections applied is not clear.

Holm-Bonferroni correction method – which is a more conservative method compared to False Discovery Rate (FDR) – was used whenever doing three or more comparisons. For example, when comparing across six ROIs as in Fig.2&3 or across four contextual features as in Fig.5. This method was the only type of correction that was applied to all the analyses, regardless of the GLM and ROI. This is now added to the Methods line 598.

12. Figure 1: integrating over the histogram of mean acceptance rates yields 24 subjects (not 25)

One subject was excluded from all behavioural and brain analyses for failing to respond frequently enough according to an a priori exclusion criterion (response rate of more than 85% or less than 15%).

This was explained in the Methods line 399 and is now additionally added to the Results line 62.

References

- Burnham, K. P., & Anderson, D. R. (2002). *Model Selection and Multimodel Inference: A Practical Information-Theoretic Approach* (2nd ed.). Springer-Verlag.
<https://doi.org/10.1007/b97636>

REVIEWER COMMENTS

Reviewer #1 (Remarks to the Author):

The authors have done a great job in revising their manuscript and dealing with the feedback from the reviewers.

Reviewer #2 (Remarks to the Author):

The authors have addressed my methodological concerns. However, I still think that the sentence (line 156-167): 'This suggests that the relationship between willingness-to-act and HB BOLD was strongest when participants received high value offers but nevertheless decided not to act...' is misleading. It is not the relationship that is strongest but rather high willingness-to-act goes together with strong HB BOLD activity in trials where participants do not act.

Reviewer #3 (Remarks to the Author):

While the authors have addressed all minor issues sufficiently, the two major flaws indicated in the first round of reviews are still present in the revised manuscript. A main concern still lies in the definition of willingness-to-act. Contrary to the authors' statement in their rebuttal letter that "the parametric design and the mixed-effect modelling of the behaviour and brain (which included both by-subject random intercepts and random slopes), strongly controlled for any across-participant variance in interest in the movie", effects of a variable 'interest-in-the-movie-clip' can only be controlled for by including trial-by-trial values of 'interest-in-the-movie-clip'. As 'interest-in-the-movie-clip' has not been recorded in this study, adequate controlling of effects is not possible. An alternative approach would be to run a small behavioural non-fMRI study using the identical paradigm and record the 'interest-in-the-movie-clip' value for each trial. This would not only yield insights on whether interest-in-the-movie-clip varies across trials, but could also demonstrate the relation between interest-in-the-movie-clip and willingness-to-act.

Regarding negative activation in MidD, it is surprising to see that a change from stimulus-onset to movement onset causes such considerable changes in the effects reported. Response times (time between offer and acceptance) are not reported in the manuscript, but may be assumed to be below one second. In combination with a temporal resolution (TR) of 2 seconds, it is questionable whether effects arising from this small temporal shift can actually be detected with the methodology presented. The radical upscaling (factor 20) used by the authors somewhat disguises the fact that all the time course plots contain only six points.

REVIEWER COMMENTS

Reviewer #1 (Remarks to the Author):

The authors have done a great job in revising their manuscript and dealing with the feedback from the reviewers.

Reviewer #2 (Remarks to the Author):

The authors have addressed my methodological concerns. However, I still think that the sentence (line 156-167): 'This suggests that the relationship between willingness-to-act and HB BOLD was strongest when participants received high value offers but nevertheless decided not to act...' is misleading. It is not the relationship that is strongest but rather high willingness-to-act goes together with strong HB BOLD activity in trials where participants do not act.

We have revised the sentence to better reflect the result. See page 9, lines 166-168:

This suggests that an increase in willingness-to-act is associated with a strong HB BOLD signal when participants received high value offers but nevertheless decided not to act, consistent with a potential role of HB in impulse control and suppression of action, when active, and release of action when less active.

Reviewer #3 (Remarks to the Author):

While the authors have addressed all minor issues sufficiently, the two major flaws indicated in the first round of reviews are still present in the revised manuscript.

1. A main concern still lies in the definition of willingness-to-act. Contrary to the authors' statement in their rebuttal letter that "the parametric design and the mixed-effect modelling of the behaviour and brain (which included both by-subject random intercepts and random slopes), strongly controlled for any across-participant variance in interest in the movie", effects of a variable 'interest-in-the-movie-clip' can only be controlled for by including trial-by-trial values of 'interest-in-the-movie-clip'. As 'interest-in-the-movie-clip' has not been recorded in this study, adequate controlling of effects is not possible. An alternative approach would be to run a small behavioural non-fMRI study using the identical paradigm and record the 'interest-in-the-movie-clip' value for each trial. This would not only yield insights on whether interest-in-the-movie-clip varies across trials, but could also demonstrate the relation between interest-in-the-movie-clip and willingness-to-act.

As requested by the reviewer we conducted a new behavioural study using the identical paradigm as the original task to control for the “interest-in-the-movie-clip” value. The study design was as follows:

Participants (N=8) performed two versions of the task, on different days. One version was exactly identical to the original design: participants were presented with offers on the screen and had to decide whether to accept an offer and engage in an effortful task for potential reward. In the second version of the task participants were asked to rate their interest in the movie scenes: at each trial, rather than being presented with offers, they were asked to rate their momentary interest in the movie on the scale of 1-9 (1=extremely uninterested; 9=extremely interested). They responded by pressing the number corresponding to their interest-rating on the keyboard. Importantly, for each participant, the timing of the “interest-in-the-movie-clip” rating question and the offer presentation exactly matched between the two versions of the task. The order of the experiments was counterbalanced between the participants. Half performed the original version followed by the rating task and the other half in reverse order.

“interest-in the-movie-clip” rating varied from trial-to-trial and from subject-to-subject, to different extents (see Figure below). However, trial by trial variation in the “interest-in the-movie-clip” did not influence participants’ decision about whether or not to act. We used three different analysis methods to show this point:

1. We created two linear mixed-effect models (similar approach as in the manuscript; analysis performed in lme4). Model 1 contained the various contextual factors in participants’ environments including reward magnitude, reward probability, and block type, which we had previously showed influence their willingness-to-act. Similar to the original model, total time was added as a covariate of no interest to account for fatigue.

Model1: $\text{logit}(\text{response}) \sim \text{rewardMagnitude} + \text{probability} + \text{blockType} + \text{totalTime}$

We replicated the main behavioural findings from the original experiment: Participants were more likely to act when offered higher magnitude rewards ($\beta=1.05\pm 0.11$, $P<0.001$), higher probability rewards ($\beta=2.39\pm 0.16$, $P<0.001$), and when they were in a poor compared to a rich block ($\beta=-0.63\pm 0.12$, $P<0.001$) (see page 6 in the manuscript).

Model 2 was similar to Model 1 but in addition contained the “interest-in-the-movie-clip” value as a predictor.

Model2: $\text{logit}(\text{response}) \sim \text{rewardMagnitude} + \text{probability} + \text{blockType} + \text{interest-in-the-movie-clip} + \text{totalTime}$

While reward magnitude, probability and block-type retained their significant effect on response (all $P_s<0.001$), we found no significant effect of “interest-in-the-movie-clip” value ($\beta=-0.07\pm 0.10$, $P=0.46$; see Figure below).

Next, we performed a formal model comparison between M1 and M2 to see whether adding “interest-in the-movie-clip” value to M1 improves the model fit: Chi-square test showed no significant improvement ($\chi^2(1)=0.54$ $P=0.46$). Additionally, we compared the

AIC scores between the two models. AIC for M1 and M2 was 725.11 and 726.58, respectively. ΔAIC 1.47 indicates that there is little to distinguish between the models.

2. We followed the same approach as in 1 but rather than using linear mixed-effect models we used a Bayesian approach (analysis performed with Bayesian Regression Models using Stan (brms) v2.15.0). Bayesian model comparison indicated substantial evidence for the null hypothesis (i.e., for Model 1) (estimated Bates factor in favour of M2 over M1: 0.32148).
3. Finally, we wondered whether the true relationship between the “interest-in the-movie-clip” value and the willingness-to-act is hidden because of some spurious correlation between the trial-by-trial variation in the “interest-in the-movie-clip” value and other factors including the reward magnitude, reward probability and block type. We therefore checked the correlation between each of these variables: we found no significant correlation between “interest-in the-movie-clip” value and reward magnitude (Pearson $r = -0.02$), reward probability ($r=0.06$), or block type ($r=0$) (see Figure below).

Together, these results suggest that even though on average participants were slightly more likely to refrain from acting when they were interested in a movie scene ($\beta=-0.07$); i) this relationship was not statistically significant; ii) “interest-in the-movie-clip” could not explain any meaningful proportion of variance in willingness-to-act as tested in a nested linear model; iii) or when using a Bayesian approach; iv) and, finally, the value was not correlated with other contextual factors in participants’ environments including reward magnitude, reward probability, and block type.

We recognise that these results are based on a small sample size. Given the Covid restrictions and the university guidelines we were extremely limited in our ability to recruit participants. Collecting data from eight participants took several weeks and entailed navigating arduous hurdles. However, the fact that we could replicate some of the main behavioural results of the experiments in this small sample gave us confidence that it is a good representative of the population.

We have now summarised these results in the Discussion lines 325-328, and in the supplementary figure 1:

a

Figure S1. Related to Figure 1. We performed a control behavioural experiment to show that interest in the movie scenes does not influence participants' decision about whether or not to act. Participants (N=8) performed two versions of the task, on different days. One version was exactly identical to the original design: participants were presented with offers on the screen and had to decide whether to accept an offer and engage in an effortful task for potential reward. In the second version of the task participants were asked to rate their interest in the movie clip: at each trial, rather than being presented with offers, they were asked to rate their momentary interest in the movie on the scale of 1-9 (1=extremely uninterested; 9=extremely interested). They responded by pressing the number corresponding to their interest-rating on the keyboard. Importantly, for each participant, the timing of the "interest-in-the-movie" rating question and the offer presentation exactly matched between the two versions of the task. The order of the experiments was counterbalanced between the participants. Half performed the original version followed by the rating task and the other half in reverse order. (a) "interest-in-the-movie-clip" rating varied from trial-to-trial and from subject-to-subject, to different extents. In the violin plots the black circles are the mean and the whiskers are the standard deviation. (b) Participants were more likely to act when offered higher magnitude rewards ($\beta=1.05\pm 0.11$, $P<0.001$), higher probability rewards ($\beta=2.39\pm 0.16$, $P<0.001$), and when they were in a poor compared to a rich block ($\beta=-0.63\pm 0.12$, $P<0.001$), replicating the main behavioural findings from the main experiment. "interest-in-the-movie-clip", however, had no significant effect on probability of acting, nor was it correlated with other contextual factors in participants' environments including reward magnitude, reward probability, and block type (c). In (b), the bars show the standardized coefficients from the linear mixed-effect model and error bars are standard error of the estimated coefficients. *** $P<0.001$.

2. Regarding negative activation in MidD, it is surprising to see that a change from stimulus-onset to movement onset causes such considerable changes in the effects reported. Response times (time between offer and acceptance) are not reported in the manuscript, but may be assuming to be below one second. In combination with a temporal resolution (TR) of 2 seconds, it is questionable whether effects arising from this small temporal shift can actually be detected with the methodology presented. The radical upscaling (factor 20) used by the authors somewhat disguises the fact that all the time course plots contain only six points.

We agree with the reviewer that the change in the reported effect might not be at first intuitive. However, there are three important points to consider:

1. The difference between the previous and the new midbrain recordings is not limited to a time difference due to the shift from stimulus to response locking. There is also a change in the voxels examined; as previously explained (but perhaps not clearly enough) we have moved from examining all of dopaminergic midbrain (MidD) to just substantia nigra pars compacta (SNc), which is known for its role in action initiation.

2. As mentioned by the reviewer the response times are quite short (mean 1.03s [min 0.5- max 2.0s]), but, importantly, they are jittered in respect to the stimulus onset. This means that even though the duration between stimulus and response onset are short, temporal locking at the response onset results in a stronger action-related BOLD signal as compared to the stimulus-locked action-related BOLD signal.

3. It is true that data from 6 TRs contribute to each trial's 10s data time course (from -2 to 8 s) but, importantly, the analysis script take into account the fact that the TRs are jittered with respect to the stimulus and response, so they are collected at a slightly different time point with respect to the experimental design on each trial. Nevertheless, we repeated the analysis, but this time did not apply any upscaling. We then compared the effect of act/no-act decision on BOLD activity in VTA and SNc at three TRs (~6s; corresponding with human hemodynamic lag) from response onset. The result showed a positive effect of act/no-act on SNc BOLD ($t(21)=2.5, p=0.02$), but not on VTA ($t(21)=1.3, p=0.19$). This confirms that the effect is not merely an artefact of upscaling.

REVIEWERS' COMMENTS

Reviewer #3 (Remarks to the Author):

The authors have now performed an additional behavioural study on the potential effects of interest-in-the-movie in the analyses. Although group size is quite small ($n=8$), the findings indicate that interest-in-the-movie is not a significant factor for the probability of acting. This important information is now also included in the manuscript.

Regarding the upscaling issue, the authors have repeated their analysis using non-upscaled data and replicated their previous findings.

Taken together, the authors have sufficiently addressed all concerns raised.